# Systemic Chemotherapy in Colorectal Peritoneal Metastases Treated with Cytoreductive Surgery: Systematic Review and Meta-Analysis

**DOI:** 10.3390/cancers16061182

**Published:** 2024-03-18

**Authors:** Marco Tonello, Carola Cenzi, Elisa Pizzolato, Riccardo Fiscon, Paola Del Bianco, Pierluigi Pilati, Antonio Sommariva

**Affiliations:** 1Surgical Oncology of Digestive Tract, Veneto Institute of Oncology IOV-IRCCS, 35128 Padua, Italy; marco.tonello@iov.veneto.it (M.T.); elisa.pizzolato@iov.veneto.it (E.P.); riccardo.fiscon@iov.veneto.it (R.F.); pierluigi.pilati@iov.veneto.it (P.P.); 2Clinical Research Unit, Veneto Institute of Oncology IOV-IRCCS, 35128 Padua, Italy; carola.cenzi@iov.veneto.it (C.C.); paola.delbianco@iov.veneto.it (P.D.B.)

**Keywords:** peritoneal metastases (PM), cytoreductive surgery (CRS), HIPEC, colorectal cancer, systemic chemotherapy (SC)

## Abstract

**Simple Summary:**

There is little evidence about the optimal timing of systemic chemotherapy (SC) in patients treated with cytoreductive surgery (CRS) and HIPEC for colorectal peritoneal metastases (CRC-PM). Due to the lack of evidence in this field, a systematic literature search and a meta-analysis of relevant studies were performed. Twenty-one studies were included in the systematic review and fifteen in the quantitative analysis (4523 patients). Postoperative SC was associated with increased overall survival compared to no SC or a preoperative SC regimen, whereas SC (pre or post) and pre-SC compared to surgery alone were not. Similar results were found for disease-free survival. Preoperative SC was not associated with an increased risk of severe surgical complications.

**Abstract:**

Background. For patients with colorectal cancer (CRC) peritoneal metastases (PM) who are eligible for cytoreductive surgery (CRS), the indication and timing of systemic chemotherapy (SC) are still under debate. This study aims to analyze the role of pre, post or perioperative SC on the survival and surgical complications of patients treated with CRS-HIPEC. Methods. After a systematic search in MEDLINE, Cochrane Database of Systematic Reviews, Scopus, Web of Science and Embase, a meta-analysis was performed to compare postoperative complications, disease-free survival (DFS) and overall survival (OS) according to SC administration and timing. PROSPERO: CRD42023478977. Results. Of 1203 studies screened, 15 were included in the meta-analysis (4523 patients). Post-operative SC was associated with increased overall survival (post-SC vs. no post-SC: HR 0.81, *p* = 0.00001, I^2^ = 0%; pre-SC vs. post-SC: HR 0.65, *p* = 0.01, I^2^ = 28%), whereas SC (pre or post) or pre-SC compared to surgery alone was not (SC vs. no SC: *p* = 0.29, I^2^ = 80%; pre-SC vs. no pre-SC: *p* = 0.59, I^2^ = 58%). Similar results were seen for DFS. SC was not associated with an increased complication rate (*p* = 0.47, I^2^ = 64%). Conclusions. Systemic chemotherapy administration in patients undergoing radical surgery for colorectal peritoneal metastases is associated with increased survival only in the adjuvant/post-operative setting. Considering the limitations of the included studies, further trials are needed to answer this unresolved question.

## 1. Introduction

Colorectal cancer (CRC) is one of the most prevalent neoplasms worldwide, and the peritoneum is the second most common site of metastasis, with approximately 2 million new cases diagnosed per year [1,2]. In comparison to other stage IV CRC patients (liver and lung), peritoneal metastases (PM) are associated with a shorter life expectancy [3].

In the past, CRC-PM was considered incurable, and palliative chemotherapy was the only appropriate option; the introduction of oxaliplatin- and irinotecan-based SC regimens combined with targeted therapy has improved the median survival of patients with CRC-PM to more than 20 months [3,4]. More recently, cytoreductive surgery (CRS) with or without hyperthermic intraperitoneal chemotherapy (HIPEC) is being offered to selected patients [5]. Surgical treatment combined with systemic chemotherapy improves overall survival, resulting in a median OS of 40–43 months [6,7].

However, even in resectable CRC-PM, the indication, optimal timing and regimen of SC are not fully established. SC can be offered as a neoadjuvant treatment prior to CRS, as adjuvant treatment after surgery, or as part of a perioperative strategy both before and after surgery [8,9,10]. Each timing of chemotherapy in patients eligible for surgery may have some advantages (treatment of undiagnosed hematogenous micrometastases, reduction of recurrence, patient selection) and disadvantages (delay in surgery, postoperative complications, side effects) [9,11].

There are only two randomized controlled trials on this topic. COMBATAC trial evaluated perioperative FOLFOX/FOLFIRI plus cetuximab versus CRS-HIPEC alone but was closed in 2014 due to insufficient accrual of patients. With obvious limitations, the results indicate the feasibility and safety of the perioperative strategy [12]. CAIRO6, which started in 2017, has a similar design, and only preliminary results are available (the perioperative arm showed similar surgical radicality and postoperative complications with a 38% major pathological response rate) [13].

At present, there is no consensus on the indication and timing of SC, and the therapeutic pathway in potentially resectable CRC-PM patients is mainly based on institutional protocols. The aim of this study is to analyze the survival outcomes and complication rates associated with SC in CRC-PM patients treated with radical surgery and HIPEC through a meta-analysis of published studies.

## 2. Materials and Methods

### 2.1. Study Design

The literature search, study design and data analysis were performed according to PRISMA (Preferred Reporting Items for Systematic Reviews and Meta-Analyses) guidelines [14]. The study has been registered (PROSPERO: CRD42023478977).

### 2.2. Search Strategy

Five medical databases (MEDLINE, Cochrane Database of Systematic Reviews, Scopus, Web of Science and Embase) were searched on 28 November 2023 for relevant studies using the following search terms [Title, Abstract]: “colorectal” AND (“hyperthermic intraperitoneal chemotherapy” OR “HIPEC”) AND (“cytoreductive surgery” OR “cytoreduction” OR “CRS”) AND (“systemic” OR “neoadjuvant” OR “adjuvant” OR “preoperative” OR “postoperative” OR “perioperative”) AND (“chemotherapy” OR “therapy” OR “treatment”).

Retrieved records were imported in EndNote X9 software (Clarivate, Philadelphia, PA, USA), which was used for the first identification of studies suitable for analysis (exclusion criteria number 1, see Section 2.3). Then, the screening process was completed by evaluation of the abstract (and, in unclear cases, the full text) of the rest of the studies.

References from selected relevant studies were manually searched to add other potentially relevant publications. 

Two researchers (M.T. and C.C.) independently selected studies from the search results according to the inclusion and exclusion criteria. Any disagreement on study inclusion between the two researchers was resolved through discussion. 

### 2.3. Selection Criteria and Outcome Measures

Inclusion criteria for this meta-analysis were: (1) patients with colorectal peritoneal metastases with pathological confirmation, treated with cytoreductive surgery (CRS) and hyperthermic intraperitoneal chemotherapy (HIPEC); (2) radical surgery, intended as completeness of cytoreduction score 0–1 rate higher than 85% of the sample; (3) no extraperitoneal disease, except for patients with radical liver surgery for hepatic metastases if the rate of such cases was less than one-third of the sample; (4) reported complete survival data: overall survival (OS) and/or disease-free survival (DFS) and/or postoperative complication rate; (5) complete data on systemic chemotherapy. Different groups of patients were created according to the use and timing of SC: administration of SC without description of timing (SC group), patients receiving SC before, after (or both) the CRS-HIPEC procedure (pre-SC, post-SC, peri-SC groups, respectively); patients treated with surgery and HIPEC alone were not included in any SC group.

Exclusion criteria were: (1) studies not suitable for analysis (duplicate articles, editorials, abstracts, book chapters, non-English papers, commentaries, letters, trial registry records; (2) studies that did not separate results according to systemic chemotherapy use and/or timing; (3) incomplete survival data; (4) studies with incomplete surgery, extraperitoneal disease, PM other than colorectal origin, study protocols, case reports, review, preclinical studies, pediatric studies.

Outcome measures were overall survival (OS), disease-free survival (DFS) and severe complication rates (Clavien-Dindo grade 3–4) following CRS and HIPEC. All relevant text, tables and figures were reviewed for data extraction.

### 2.4. Data Extraction

Data were extracted only from original articles, using a pro forma with a set of predefined parameters: first author, year of publication, journal, study design, number of participating centers, nationality, number of patients, postoperative complications, timing of systemic chemotherapy (pre, post, peri-SC) or use of systemic chemotherapy (SC vs. no SC), median OS and DFS, rate of radical surgery, rate of extraperitoneal (hepatic) metastases, systemic chemotherapeutic agents used (if reported), HIPEC protocol.

### 2.5. Statistical Analysis

All meta-analyses were performed with Review Manager 5.3 (Cochrane Collaboration, Nordic Cochrane Centre, Copenhagen, Denmark) using the inverse variance of the log of the hazard ratio (HR) or odds ratio (OR) of events (Mantel-Haenszel test, an inferential test for the association between two binary variables, while controlling for a third confounding nominal variable), where appropriate. A *p*-value of 0.05 or less was considered significant. 

Overall survival (OS) and disease-free survival (DFS), hazard ratios (HR) and confidence intervals (CI) were extracted from the included studies. In the case of incomplete or partially reported data, hazard ratios (HR) for time-to-event outcomes with 95% confidence intervals (95%CI) were estimated using methods suggested by Tierney [15] and Hebert [16]; these methods allow extraction or estimation of the hazard ratio to standardize time-to-event outcomes. For incomplete data on mean/median survival and survival ranges, missing values were estimated using the method of Hozo [17], modified by Wan [18]; this method provides an estimation of mean and standard deviation using study available data, such as sample size or range of values.

The I^2^ statistic was used to determine the heterogeneity of the included studies. I^2^ values of 25–49%, 50–74% and above 75% were considered low, moderate and high heterogeneity levels, respectively [19]. When high heterogeneity was present, a sub-analysis to determine confounding factors among studies was performed with meta-regression analysis (Open MEE software, http://www.cebm.brown.edu/openmee/index.html (accessed on 6 March 2024)) [20]. In case of impossibility to determine and solve potential biases, a random-effects model was used. Otherwise, when the I^2^ statistic was less than 50% (low or moderate), the fixed-effects model was used. 

Forest and funnel plots were used to graphically present the statistical results. 

### 2.6. Quality Assessment of Retrieved Articles

Two researchers (M.T. and C.C.) independently assessed the quality of the articles using a quality assessment list based on the Newcastle–Ottawa Quality Assessment Scale (NOS) (Appendix A). In case of disagreement, the most conservative (lower) value was used.

## 3. Results

### 3.1. Study and Patient Characteristics

After the literature search, 2442 records were retrieved. The initial assessment, conducted on Title and Abstract fields, excluded 1810 records (1683 duplicates and 127 abstracts, book chapters, comments/letters, dissertations/thesis, non-English, trial registry records), resulting in 632 potential studies. The second round of screening with full-text analysis excluded another 611 studies (reviews, case reports, guidelines, pediatric or preclinical studies, study protocols, studies not reporting survival outcomes or systemic chemotherapy regimen/timing, studies including non-CRC PM or other locoregional treatment than HIPEC). In the end, 21 relevant articles reporting at least one of the selected outcomes were included in the systematic review. These articles were extensively reviewed, and six were excluded due to a high rate of incomplete surgery, CC2 > 25% (Appendix A); finally, 15 studies (4523 patients) were included in the quantitative analysis (Figure 1). Of these, 14 were case series (one prospective cohort study [21] and 13 were retrospective, of which 11 were cohort studies [5,9,22,23,24,25,26,27,28,29,30] and two were case-control studies with propensity score analysis [31,32]) and one was a randomized controlled trial [13]. Four studies were monocentric, and 11 were multicentric; the median year of patient enrollment was 2009 (IQR range 2007–2012). Study characteristics and details are summarized in Table 1, while survival and complication data are summarized in Table 2. 

For survival analysis (OS and DFS), the studies were grouped into four sub-analyses: (1) systemic chemotherapy administration (at any time) compared to patients receiving CRS-HIPEC only (SC vs. no SC, four studies); (2) patients receiving preoperative SC compared to patients treated with upfront CRS-HIPEC, regardless of whether they also received postoperative SC (pre-SC vs. no pre-SC, 7sevenstudies); (3) patients receiving postoperative SC compared to patients treated with CRS-HIPEC, regardless of whether they also received preoperative SC (post-SC vs. no post-SC, seven studies); (4) patients receiving preoperative SC without postoperative SC compared to patients receiving postoperative SC without preoperative SC (pre-SC vs. post-SC, four studies). Some studies were included in more than one sub-group, according to the reported data.

### 3.2. Outcome Measures

#### 3.2.1. Overall Survival

Twelve studies reported overall survival data. Postoperative systemic chemotherapy was associated with improved survival with low inter-study heterogeneity (post-SC vs. no post SC: OS HR 0.81, 95%CI 0.73–0.9, *p* = 0.00001, I^2^ = 0%; post-SC vs. pre-SC: OS HR 0.65, 95%CI 0.46–0.91, *p* = 0.01, I^2^ = 28%). Patients receiving SC (before or after surgery) had similar survival compared to patients treated with CRS-HIPEC only (SC vs. no SC: OS HR 0.73, 95%CI 0.40–1.32, *p* = 0.29, I^2^ = 80%). Patients who received systemic chemotherapy before surgery also had similar survival to those who did not (pre-SC vs. no pre-SC: OS HR 0.95, 95%CI 0.79–1.14, *p* = 0.59, I^2^ = 58%) (Figure 2). 

#### 3.2.2. Disease-Free Survival

Seven studies reported disease-free survival data. Also considering DFS, a significant improvement in survival was observed only in patients who received postoperative systemic chemotherapy compared to patients who did not (post-SC vs. no post-SC: DFS HR 0.82, 95%CI 0.72–0.94, *p* = 0.003, I^2^ = 0%). Patients receiving SC after surgery tended to have better DFS compared to patients who received preoperative SC (post-SC vs. pre-SC: DFS HR 0.75, 95%CI 0.54–1.04, *p* = 0.08, I^2^ = 52%). All other comparisons showed similar DFS between groups, with moderate/high inter-study heterogeneity (SC vs. no SC: DFS HR 0.85, 95%CI 0.5–1.46, *p* = 0.56, I^2^ = 81%; pre-SC vs. no pre-SC: DFS HR 1.22, 95%CI 0.81–1.83, *p* = 0.35, I^2^ = 87%) (Figure 3).

#### 3.2.3. Postoperative Complications

Postoperative complications were reported in eight studies. The addition of preoperative systemic chemotherapy was not associated with an increased risk of major complications (Clavien-Dindo grade 3–4), although the heterogeneity was moderate (complication rate with pre/peri-SC 24.8% vs. complication rate without pre/peri-SC 24.1%; HR 1.16, 95%CI 0.78–1.79, *p* = 0.47, I^2^ = 64%) (Figure 4).

### 3.3. Sensitivity Analysis and Publication Bias

The heterogeneity of the included studies, reported using the I^2^ statistic, was low in the analysis of OS and DFS of postoperative systemic chemotherapy (0% and 28%), whereas it was moderate in studies comparing SC with pre-SC and postoperative complications (OS pre-SC vs. no pre-SC: 58% and DFS pre vs. post-SC: 54%; complications: 64%) and high in the remaining studies (OS SC vs. no SC: 80%; DFS SC vs. no SC: 81%, pre-SC vs. no pre-SC: 87%). 

Meta-regression was performed to determine confounding factors in the postoperative complications sub-analysis, but was not applicable to other high heterogeneity sub-analyses due to the lack of statistical reliability when fewer studies are included. Nevertheless, it was not possible to identify the cause of high I^2,^ despite using meta-regression analysis to test confounding factors such as year of patients’ enrollment, study design or use of target therapy in studies reporting complications. Grouping studies for the administration of triplet systemic chemotherapy (FOLFOXIRI), even though not statistically relevant (*p* 0.196), was the only factor able to reduce study heterogeneity by 30% (residual I^2^ 49.4%, R^2^ 29.8%); complete results are in the Appendix A. Therefore, given the impossibility of determining and solving potential biases in included studies, random-effect analyses were performed in comparisons with I^2^ > 50%.

Publication quality was assessed as good using the Newcastle–Ottawa Quality Assessment Scale (NOS) (median NOS score 7/9 for cohort and 8/9 for case-control studies) (full data in Appendix A). In addition, funnel plots of the analyses performed are provided in the Appendix A.

## 4. Discussion

At present, there is robust evidence that selected patients with limited peritoneal metastases (PM) of colorectal cancer (CRC) can be effectively treated with surgery (CRS ± HIPEC) and achieve long-term survival (approximately 40–45 months) [33,34].

The use of systemic chemotherapy (SC) and its timing in this sub-group of CRC patients has not been fully elucidated; indeed, optimal chemotherapy regimens and strategies are still a topic of debate [35]. In the absence of shared guidelines, centers with PM expertise have adopted different approaches: administering SC before surgery (preoperative schedule/neoadjuvant), dividing the number of cycles planned both before and after surgery (perioperative), or treating patients with upfront surgery, possibly followed by adjuvant chemotherapy (postoperative). 

Each strategy (pre, peri or postoperative SC) has advantages and disadvantages. Preoperative chemotherapy could reduce the burden of peritoneal disease, increase the completeness of cytoreduction and limit the extension of surgery, with a potential reduction in the complication rate [9,36]. However, it has been shown that patients considered unresectable at laparoscopic exploration before SC are very unlikely to be converted to radical surgery (CC0) after systemic treatment [37], and the objective response rate (radiological or histological) after SC is estimated to be up to 50% of patients [9,36]. Preoperative SC could also treat hematogenous micrometastases, thereby reducing the risk of extraperitoneal recurrence [38,39]. These potential advantages may be lost with postoperative SC; in addition, as cytoreductive surgery is a relatively morbid procedure, there is a risk that chemotherapy may not be administered in the event of severe complications, delaying recovery after surgery [40].

On the other hand, preoperative SC may be associated with an increased surgical complication rate [11,12,41,42] or may lead to patient exclusion (undertreatment) in the event of toxicity or poor response that precludes access to surgery in potentially resectable patients (e.g., peritoneal progression in a still “resectable” patient after preoperative SC) [43,44].

In fact, there are some retrospective reports in the scientific literature with controversial results regarding the potential benefit of pre- or postoperative SC in CRS-HIPEC. Three old series described an improvement in OS in the postoperative setting [10,45], while four more recent studies have favored pre-operative SC (but only one showed a survival advantage in multivariate analysis) [9,23,25,27,46].

Another proposed approach is to split chemotherapy before and after surgery (perioperative SC). A perioperative approach (before and after CRS) has the advantage of assessing the histological response to preoperative SC, which provides important information on the chemosensitivity of a tumor and the possibility of changing the drug regimen in case of histologically proven chemoresistance. Two randomized controlled trials (COMBATAC and CAIRO6) comparing perioperative SC with surgery alone demonstrated the feasibility and safety of this dosing regimen, and the latter also reported an interestingly high (38%) major pathological response rate [12,13]. While the COMBATAC trial was closed due to low accrual, survival results from CAIRO6 are expected in the next few years.

Two independent systematic reviews conducted in 2017 failed to demonstrate a clear advantage of specific timing of SC administration: the first study suggested a potential advantage of pre- and perioperative regimens [47], while the second reported a weak association between postoperative SC and longer OS, especially after incomplete cytoreduction) [8]. Two very recent studies published in 2023 reported a survival advantage with post-operative SC, through propensity score analysis in large retrospective series [31,32].

The results of this meta-analysis confirmed better outcomes (overall survival and disease-free survival) when SC is administered in the postoperative period after complete cytoreductive surgery and HIPEC. Indeed, patients who received postoperative systemic chemotherapy had a reduced risk of cancer-related death and recurrence compared to patients who did not receive systemic chemotherapy after surgery (post-SC vs. no post-SC: OS HR 0.81, *p* 0.0001; DFS HR 0.82, *p* 0.003). Similar results were seen in patients treated with postoperative SC compared to preoperative SC (OS HR 0.65, *p* 0.01; DFS HR 0.75, *p* 0.08). The results obtained for both sub-groups were robust with low inter-study heterogeneity (I^2^ = 0%, I^2^ = 0% and I^2^ = 28%, moderate only for pre-SC vs. post-SC, I^2^ = 52%). With regard to preoperative and perioperative systemic chemotherapy regimens, the present results do not allow a definitive conclusion, as the lack of survival improvement (*p* 0.59 and 0.35) cannot be assured due to the high heterogeneity of the studies (I^2^ = 58% and 87%). Further evaluations for biases in included studies were not possible (e.g., meta-regression) due to the reduced number and type of the included studies (mainly retrospective series) in each comparison. In line with a recent randomized controlled trial, severe postoperative complications were not higher in pre-treated patients (*p* 0.47), even though these results should be interpreted with caution due to the moderate inter-study heterogeneity (I^2^ = 64%). Sub-group analysis with meta-regression showed a potential bias when patients are receiving triplet systemic chemotherapy (FOLFOXIRI). This aspect should be further evaluated, since the lack of a strong evidence in our analysis.

The reasons for the improved survival in the postoperative setting are yet to be fully determined. Several patient- and tumor-related factors may have influenced the decision on the most appropriate SC strategy in CRC-PM patients selected for surgery. One possibility is that patients treated with upfront surgery and postoperative SC may have a better functional status, low-risk pathological features or lower disease burden compared to patients who received pre-operative SC, in a “conversion” or “downstaging” approach. Furthermore, considering surgery and systemic chemotherapy as a single multimodal treatment, estimating survival from the date of surgery (as reported in all the studies analyzed) could have biased the results. In fact, the group of patients treated postoperatively started their follow-up period earlier than patients who underwent SC before surgery. 

One of the main limitations of this study is the retrospective design of the included articles, although the assessed quality of the included studies was good. The lack of intention to treat survival data partially limits the results of the analysis, as it is likely that a significant number of patients treated before surgery were excluded from CRS due to disease progression (peritoneal or systemic) or SC toxicity. Moreover, it is reasonable to assume that some patients in the postoperative setting were excluded from SC because of postoperative complications or for the need for a long rehabilitation period after surgery. There is only one ongoing randomized controlled trial with no survival results, which was only included due to the availability of data in the postoperative complications sub-group [30]. Another limitation of survival analysis in CRC stage IV patients is related to iterative treatments, including different lines of systemic chemotherapy or repeated CRS at recurrence. Therefore, OS is the end result of a multimodal and iterative approach that may mask the true effect of systemic chemotherapy administered close to CRS-HIPEC. Despite these limitations, this study incorporates the most recently published studies with stringent surgical selection criteria (completeness of cytoreduction, reduced number of synchronous liver metastases) of peritoneal-only stage IV patients with CRC-PM. In addition, the results of postoperative systemic chemotherapy are robust with low inter-study heterogeneity. Nevertheless, new prospective studies or randomized trials are needed to determine the role of systemic chemotherapy and the optimal regimens in CRC-PM patients eligible for cytoreductive surgery.

## 5. Conclusions

In CRC-PM patients eligible for CRS, the administration of systemic chemotherapy remains a topic of debate without shared guidelines on timing and protocols. Using published data, it appears that postoperative systemic chemotherapy would be associated with a survival benefit, whereas the role of pre- or perioperative timing remains controversial. Given study limitations, additional randomized trials are needed to define the role and timing of systemic chemotherapy in this subset of patients.

## Figures and Tables

**Figure 1 cancers-16-01182-f001:**
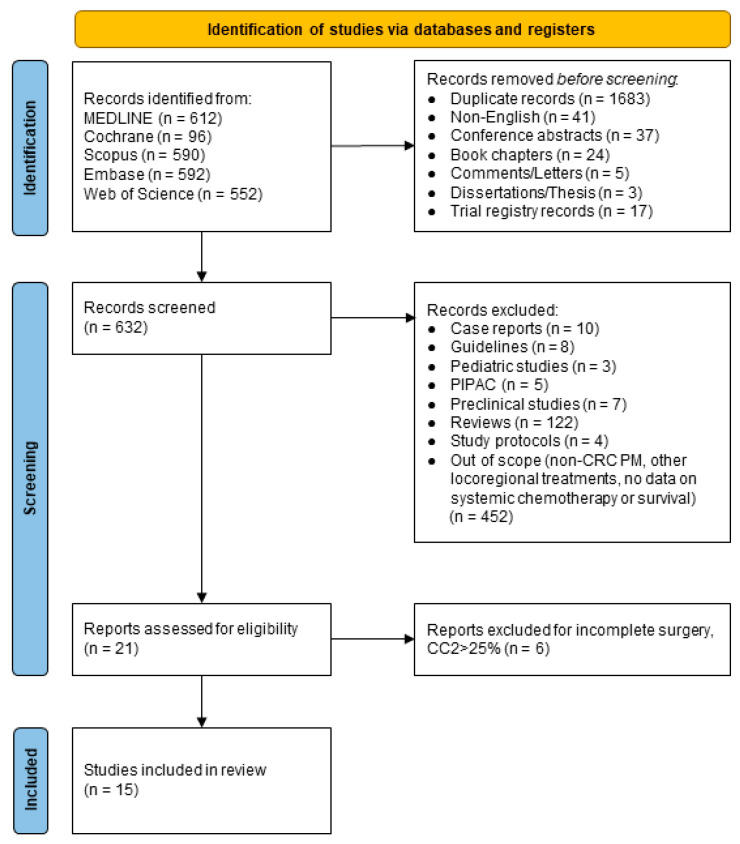
PRISMA flowchart of the literature selection process.

**Figure 2 cancers-16-01182-f002:**
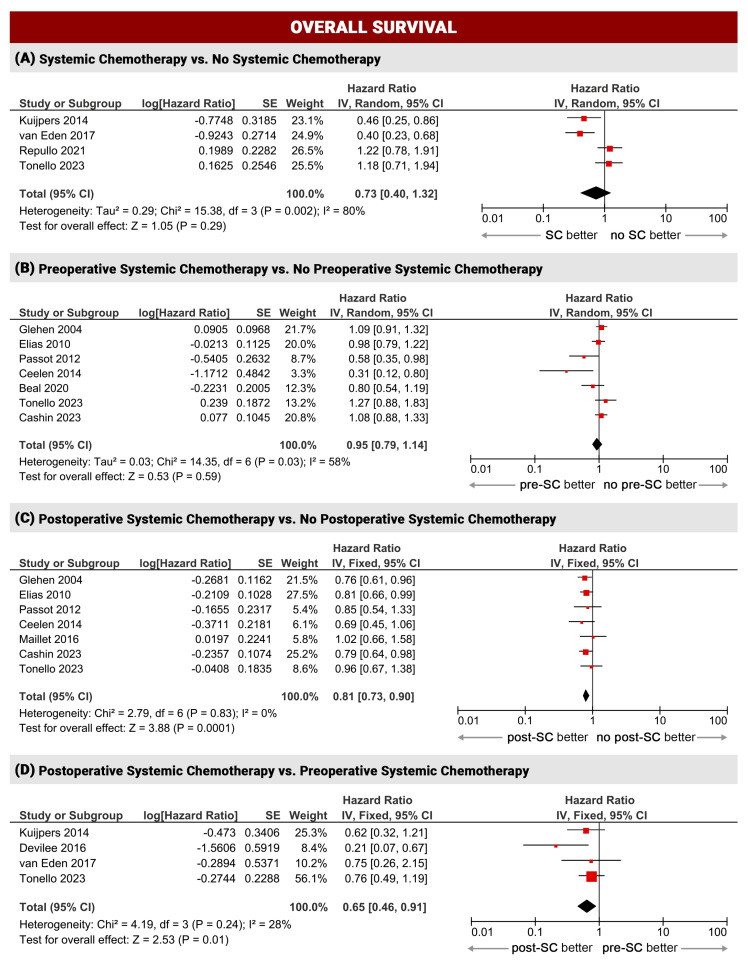
Overall survival meta-analysis [5,9,21,22,23,24,25,26,27,28,31,32].

**Figure 3 cancers-16-01182-f003:**
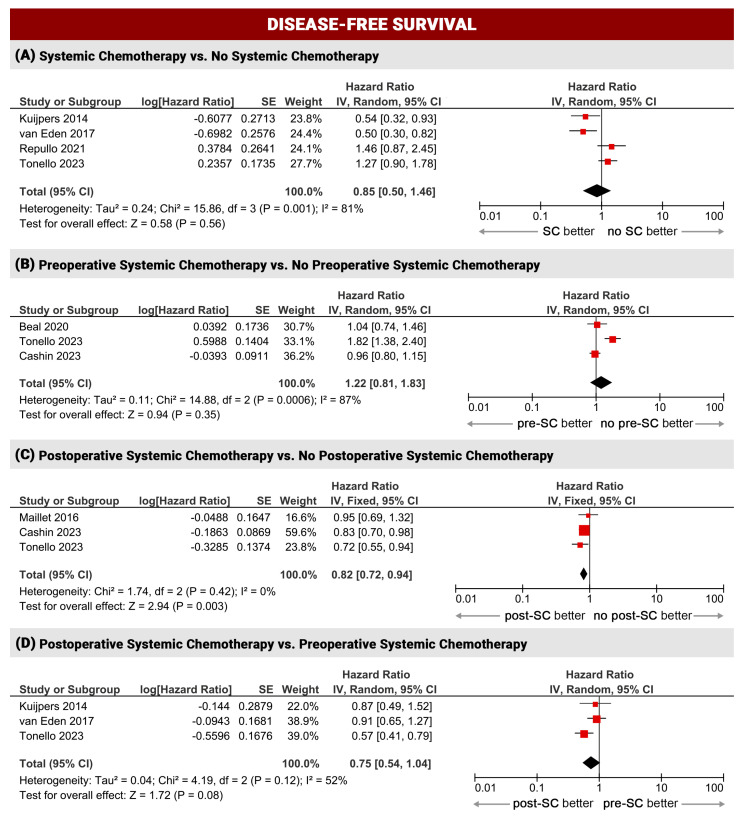
Disease-free survival meta-analysis [21,24,26,27,28,31,32].

**Figure 4 cancers-16-01182-f004:**
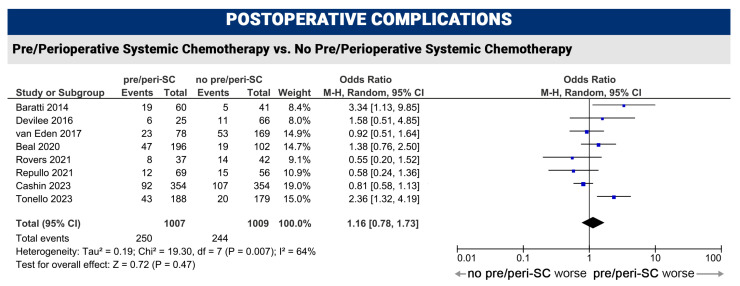
Severe postoperative complications (grade 3–4 according to Clavien-Dindo classification) meta-analysis [13,25,26,27,28,30,31,32].

**Table 1 cancers-16-01182-t001:** Characteristics of included studies.

First Author, Year	Study Design (Centers n)	Study Period, Country	Sample Size (n)	PCI	CC0-1 (%)	Liver M+ (%)	SC Regimens	Target Therapy (%)	HIPEC Regimen
Glehen, 2004 [5]	retrospective multicentric (28)	1987–2002 Worldwide	506	n.r.	75	12	FOLFOX, FOLFIRI, FOLFOXIRI	n.r.	MMC, CIS-MMC, OX
Elias, 2010 [22]	retrospective multicentric (23)	1990–2007 Belgium, Canada, France, Switzerland	523	10	95	15	FOLFOX, FOLFIRI, FOLFOXIRI	0	MMC, CIS-MMC, OX (±IRI)
Passot, 2012 [9]	retrospective single center	1991–2010 France	120	8	86	0	FOLFOX, FOLFIRI	19	MMC, MMC-OX, MMC-IRI
Baratti, 2014 [30]	retrospective multicentric (2)	2004–2012 Italy	101	10	98	8	CAPOX, FOLFOX	22	CIS-MMC
Ceelen, 2014 [23]	retrospective single center	2002–2012 Belgium	166	n.r.	87	n.r.	FOLFOX, FOLFIRI	42	MMC, OX
Kuijpers, 2014 [24]	retrospective single center	2004–2012 The Netherlands	71	n.r.	100	n.r.	CAPOX, FOLFOX	n.r.	MMC
Devilee, 2016 [25]	retrospective single center	2007–2014 The Netherlands	91		100	20	CAPOX, FOLFOX	28	n.r.
Maillet, 2016 [21]	prospective multicenter (4)	2004–2012 France	231	9	100	5	5-FU, FOLFOX, FOLFIRI	34	OX, OX-IRI, MMC
van Eden, 2017 [26]	retrospective single center	2004–2015 The Netherlands	280	n.r.	100	6	CAPOX, FOLFOX	n.r.	MMC, OX
Beal, 2020 [27]	retrospective multicentric (12)	2000–2017 USA	298	13	88	0	CAPOX, FOLFOX FOLFIRI, FOLFOXIRI	54	MMC, OX
Repullo, 2021 [28]	retrospective multicentric (2)	2008–2017 Belgium	125	6	100	30	FOLFOX, FOLFIRI, 5-FU-CAPE	58	MMC, OX
Rovers, 2021 [13]	RCT (9)	2012–2017 The Netherlands	79	9	87	0	CAPOX, FOLFIRI, FOLFOX	98	MMC, OX
Hanna, 2022 [29]	retrospective multicentric (2)	2011–2019 USA	79	11	93	17	FOLFOX, FOLFIRI, CAPOX	63	MMC
Cashin, 2023 [31]	retrospective multicentric (39)	1991–2018 Worldwide	1486	10	99	13	n.r.	n.r.	CIS, IRI, MMC, OX-IRI
Tonello, 2023 [32]	retrospective multicentric (13)	1997–2017 Italy	367	9	100	0	FOLFOX, FOLFIRI, FOLFOXIRI	58	CIS-MMC, OX

Abbreviations. RCT: randomized controlled trial; CAPOX: capecitabine plus oxaliplatin; FOLFIRI: folinic acid, fluorouracil plus irinotecan; FOLFOX: folinic acid, fluorouracil plus oxaliplatin; FOLFOXIRI: folinic acid, fluorouracil, plus oxaliplatin plus irinotecan; MMC: mitomycin C; CIS: cisplatin; OX: oxaliplatin; IRI: irinotecan; PCI: Peritoneal Cancer Index; SC: Systemic Chemotherapy; CC: Completeness of Cytoreduction; n.r.: not reported.

**Table 2 cancers-16-01182-t002:** Patients’ distribution and reported survivals according to systemic chemotherapy administration and timing.

	SC				No SC				Pre-SC				Post-SC				Peri-SC				No Pre-SC				No Post-SC			
	n	OS	DFS	C	n	OS	DFS	C	n	OS	DFS	C	n	OS	DFS	C	n	OS	DFS	C	n	OS	DFS	C	n	OS	DFS	C
Glehen, 2004 [5]	-	-	-	-	-	-	-	-	275	19	-	-	204	25	-	-	-	-	-	-	231	20	-	-	302	16	-	-
Elias, 2010 [22]	-	-	-	-	-	-	-	-	370	30	-	-	232	31	-	-	-	-	-	-	153	30	-	-	291	27	-	-
Passot, 2012 [9]	-	-	-	-	-	-	-	-	90	37	-	-	77	36	-	-	-	-	-	-	30	24	-	-	43	14	-	-
Baratti, 2014 [30]	-	-	-	-	-	-	-	-	60	-	-	19	-	-	-	-	-	-	-	-	41	-	-	5	-	-	-	-
Ceelen, 2014 [23]	-	-	-	-	-	-	-	-	61	29	-	-	83	30	-	-	-	-	-	-	105	25	-	-	83	22	-	-
Kuijpers, 2014 [24]	55	3	15	-	16	14	4	-	25	27	13	-	32	24	14	-	-	-	-	-	-	-	-	-	-	-	-	-
Devilee, 2016 [25]	-	-	-	-	-	-	-	-	-	-	-	-	66	39	-	11	25	n.e.	-	6	25	-	-	-	-	-	-	-
Maillet, 2016 [21]	-	-	-	-	-	-	-	-	-	-	-	-	151	43	13	42	-	-	-	-	-	-	-	-	70	50	10	25
van Eden, 2017 [26]	247	41	22	76	33	34	17	8	-	-	-	-	169	43	22	53	78	37	20	23	-	-	-	-	-	-	-	-
Beal, 2020 [27]	-	-	-	-	-	-	-	-	196	33	14	47	-	-	-	-	-	-	-	-	102	22	13	19	-	-	-	-
Repullo, 2021 [28]	-	-	-	-	56	72	17	15	-	-	-	-	-	-	-	-	69	43	11	12	-	-	-	-	-	-	-	-
Rovers, 2021 [13]	-	-	-	-	-	-	-	-	37	-	-	8	-	-	-	-	-	-	-	-	42	-	-	14	-	-	-	-
Hanna, 2022 [29]	-	-	-	-	-	-	-	-	34	78	30	-	-	-	-	-	45	61	12	-	-	-	-	-	-	-	-	-
Cashin, 2023 [31]	-	-	-	-	-	-	-	-	354	35	12	92	389	46	13	143	-	-	-	-	354	37	13	107	389	37	11	159
Tonello, 2023 [32]	294	38	13	55	73	55	18	8	119	36	9	32	106	43	16	12	69	38	14	11	179	51	17	20	192	39	12	-

Abbreviations. n: number of patients; OS: median overall survival (months); DFS: median disease-free survival (months); C: severe postoperative complications (grade 3–4 according to Clavien-Dindo classification, number of events); SC: systemic chemotherapy; Pre: preoperative SC; Post: postoperative SC; Peri: perioperative SC; n.e.: not estimable.

## Data Availability

Data sharing is not applicable to this article since no new data were created.

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
