# Peer review of "Systemic Chemotherapy in Colorectal Peritoneal Metastases Treated with Cytoreductive Surgery: Systematic Review and Meta-Analysis"

_cancers, 2024, doi:10.3390/cancers16061182_

Round 1

Reviewer 1 Report

Comments and Suggestions for Authors

The study presented in the manuscript systematically reviews and meta-analyzes the effect of systemic chemotherapy (SC) on survival outcomes in patients with colorectal peritoneal metastases (CRC-PM) undergoing cytoreductive surgery (CRS) and hyperthermic intraperitoneal chemotherapy (HIPEC). The findings suggest a clear benefit of postoperative systemic chemotherapy in patients undergoing CRS and HIPEC for CRC-PM, highlighting its role in improving survival outcomes without significantly increasing the risk of severe surgical complications. However, I have doubts about whether this study can contribute to an already existing review (https://link.springer.com/article/10.1245/s10434-022-11699-7). While the manuscript addresses an important clinical question, significant improvements in methodology, analysis, and discussion are required to meet the publication standards of the journal.

1- Please explain why, despite mentioning this article (Ref 45. Zhou, S.; Jiang, Y.; Liang, J.; Pei, W.; Zhou, Z. Neoadjuvant Chemotherapy Followed by Hyperthermic Intraperitoneal Chemo-474 therapy for Patients with Colorectal Peritoneal Metastasis: A Retrospective Study of Its Safety and Efficacy. World J Surg Oncol 475 2021, 19, 151, doi:10.1186/s12957-021-02255-w. ) in the manuscript, you did not include it in the tables and analyses.

2- When searching with the search terms you specified, especially in WoS and Embase, more studies appear, which raises concerns about reliability.

3- It is not clear exactly how the authors narrowed the number of articles. Please elaborate on the exclusion criteria (for example book chapters, books, case reports, editorial letters, review articles, retrospective studies, single-arm studies, and opinion papers; animal studies; studies not in English) with numbers.

4- The manuscript reports heterogeneity in some comparisons but does not sufficiently explore its sources or address it through subgroup analyses or meta-regression. This oversight limits the ability to understand the variability in study outcomes and to generalize the findings.

Comments on the Quality of English Language

Moderate editing of English language required

Reviewer 2 Report

Comments and Suggestions for Authors

In this manuscript, Tonello et. al. have explored the current state of the literature and performed a meta-analysis on CRC metastasis and its treatments, including chemotherapy. This is an interesting manuscript with important clinical output. However, there are several areas that need improvement:

  1. The manuscript is well written and very informative. but most of the sections are not adequately expanded. All the sections need to be expanded, for instance, introduction, and the support of observations needs to be supported by recent literature.
  2. The authors need to state inclusion and exclusion criteria in more detail. Also, the authors need to expand on the methodology used to remove duplicate studies.
  3. The authors need to expand and improve the introduction section. For instance: CRC and its current clinical framework need to be expanded to provide a better context for their study.
  4. The authors need to significantly expand the statistical analysis section. Instead of references, the authors can provide details of the methodology, and principles behind it. Most importantly, why these tests were selected and other methods were not, needs to be expanded.
  5. conclusion should crystallize the clinical outlook and framework for the readers. Therefore, the authors need to expand this segment.
Comments on the Quality of English Language

The English needs revision for flow.

Reviewer 3 Report

Comments and Suggestions for Authors

This a well-conducted meta-analysis on a relatively narrow topic, which can have some relevance in the field.

Some comments, with a constructive intent:

- the 'simple summary' is not so simple, but rather a shorter abstract. please revise.

- Abstract, line 20: the word 'and' between CRC and PM is likely missing.

- can the Authors expand on the new lines of research? are they sure that randomized trials can ever be conducted? any alternative?

Comments on the Quality of English Language

Moderate English polishing is required.

Round 2

Reviewer 1 Report

Comments and Suggestions for Authors

I am satisfied that the authors have addressed all of my previous concerns about the article. It is now much improved and I feel that it is now suitable for publication.

Comments on the Quality of English Language

Minor editing of English language required